# Photoreactions of the C_2_H_4_–SO_2_ Complex in a Low-Temperature Matrix Investigated by Infrared Spectroscopy and Density Functional Theory Calculations

**DOI:** 10.3390/molecules29225362

**Published:** 2024-11-14

**Authors:** Taito Takahashi, Fumiyuki Ito, Jun Miyazaki

**Affiliations:** 1Department of Natural Sciences, School of Engineering, Tokyo Denki University, 5 Senju-Asahi-cho, Adachi-ku 120-8551, Tokyo, Japan; 18es041@ms.dendai.ac.jp (T.T.); jmiya@mail.dendai.ac.jp (J.M.); 2National Institute of Advanced Industrial Science and Technology (AIST), Onogawa 16-1, Tsukuba 305-8569, Ibaraki, Japan

**Keywords:** ethylene, sulfur dioxide, photolysis, matrix isolation, infrared spectroscopy, density functional calculation

## Abstract

Ethylene and sulfur dioxide molecules were co-deposited on a CsI window at cryogenic temperature, and the photoproducts upon UV irradiation were observed using Fourier transform infrared (FTIR) spectroscopy. The products were found to be UV wavelength-dependent; at shorter wavelengths (λ = 266 nm) one strong peak was observed while more than three peaks were identified at longer UV wavelengths (λ = 300 nm). Spectral features changed seamlessly along with UV wavelength. Density functional theory (DFT) calculations were carried out for potential products, and spectral matches between observations and calculations seemed satisfactory, assuming a cyclic molecule (oxathietane 2-oxide) as the main photoproduct at longer UV wavelengths. On the other hand, the spectra of photoproducts at shorter UV wavelengths were reproduced by assuming the decomposition products of an intermediate, from the supplementary experiments using deuterated samples. Plausible photoreaction schemes were presented to account for the observed photoproducts.

## 1. Introduction

It is known that the oxidation of volatile organic compounds (VOCs) contributes to the secondary organic aerosols (SOAs) in the atmosphere. The main oxidant species in these processes involve ozone (O_3_), hydroxy (OH) and nitrate (NO_3_) radicals. On the other hand, various organosulfate (OS) compounds have been identified in the atmosphere, suggesting that gas phase sulfur dioxide (SO_2_) serves as a reactant for OS formation [1,2]. The reactions of unsaturated organic compounds with SO_2_ have been studied for years [3,4,5,6]; polymer formation from olefins and SO_2_ in the condensed phase was first reported in 1898. Dainton and Ivin observed photochemical reactions of SO_2_ and olefins in the gas phase for the first time [6]. They proposed an addition mechanism of electronically excited SO_2_ to the C=C double bond followed by sulfinic acid formation. They measured the UV spectra of the mixture of SO_2_ and olefins in the gas phase and suggested a strong charge–transfer interaction between them [7]. Since then, analogous systems have been investigated extensively [8,9,10,11,12]. Most of the photochemical reactions have been studied at ambient temperature, and the primary step of the prototypical system C_2_H_4_–SO_2_ has not been clarified yet, to the best of our knowledge. In this respect, a photochemical study of the molecular complex C_2_H_4_–SO_2_ is crucial to elucidating the details of reaction mechanisms. The molecular complex C_2_H_4_–SO_2_ was observed in the gas phase by FTMW technique [13,14] and IR spectroscopy in matrices [15], and those results were compared with theoretical calculations [16,17,18]. It has a slipped parallel structure in an electronic ground state. Makarov et al. reported photolysis of the C_2_H_4_–SO_2_ complex isolated in a supersonic jet by the UV resonance multiphoton ionization technique [19]. They successfully obtained rotationally resolved action spectra of the C_2_H_4_–SO_2_ complex and found that the two molecular planes of SO_2_ and C_2_H_4_ became closer upon photoexcitation. Such structural change would enhance the overlap of the π orbitals of the two molecules to lead to the photoreaction. Quite recently Salta et al. reported a theoretical study of C_2_H_4_–SO_2_ cycloaddition reactions and identified three cyclic molecules and subsequent decomposition products [20]. Their results are, however, based on the singlet potential energy surface and are not directly correlated to the photochemistry of the C_2_H_4_–SO_2_ system. Therefore, it is not clear whether the cycloaddition reactions occur in a concerted manner through the overlapped π orbitals upon photoexcitation.

In the present study, we explored the primary step of the photoreaction of the C_2_H_4_–SO_2_ complex in a cryogenic matrix using infrared spectroscopy. Spectral signatures of the photoproducts after UV irradiation have been compared with theoretical simulations, similarly to our previous studies [21].

## 2. Materials and Methods

### 2.1. Experiments

An SO_2_ gas diluted by Ar (1:100) and gaseous ethylene were purchased from Tomoe Shokai (Ohta-ku, Japan) and GL Sciences (Shinjuku-ku, Japan), respectively, and used without further purification. C_2_H_4_ was diluted with Ar at various ratios. The SO_2_/Ar sample was mixed with additional Ar to obtain more dilute samples. A typical concentration of SO_2_/Ar and C_2_H_4_/Ar samples was guest: Ar = 1:760. The two samples were co-deposited onto a CsI window separately through the twin-valve inlets [21]. The window temperature was maintained at 10–30 K using a circulating helium refrigerator. Matrix-isolated species were irradiated by a tunable UV light source obtained by the second harmonic generation (SHG) of a pulsed dye laser (ND6000, Continuum, Dallas, TX, USA) or the fourth harmonic generation (FHG) of Nd: YAG laser (Surelite-II, Continuum, Dallas, TX, USA) at 266 nm. Typical fluences were 0.6~2 mJ/pulse for the SHG at 280–310 nm and 10 mJ/pulse for the FHG. The wavelength region corresponded to the S_1_-S_0_ transition of SO_2_, the chromophore of the C_2_H_4_–SO_2_ complex. Infrared spectra before and after the irradiation were recorded with an FT/IR-6100 spectrometer (JASCO, Hachioji, Japan) to obtain the difference spectra for each species. Details of the instrumentation have been described in our previous articles [21,22].

The spectra thus obtained are presented in Figure 1. In the first stage of the assignment of photoproducts, we referred to online databases (NIST Chemistry WebBook, PubChem) as well as a search engine (Google Scholar) to explore the vibrational spectra of the potential molecules of the type C_k_H_l_O_m_S_n_ (k ≤ 2, l ≤ 4, m ≤ 2, n ≤ 1). Most of the candidate molecules do not have reported infrared data. We, therefore, carried out extensive simulations of vibrational peaks theoretically, as described below.

### 2.2. Calculations

For the assignment of photoproducts, density functional theory (DFT) calculations were carried out for candidate molecules with the formula C_k_H_l_O_m_S_n_ (k ≤ 2, l ≤ 4, m ≤ 2, n ≤ 1). For the C_2_H_4_SO_2_ molecules, 69 isomeric structures were constructed manually using GaussView 5.0 and optimized at the semi-empirical PM6 level, followed by further optimization at the B3LYP/6-31+G(d) level. Finally, the top 14 stable isomers were reoptimized at B3LYP/cc-pVTZ. For smaller species, 42 molecules were subject to structure optimization at the B3LYP/6-31G(d) levels of theory. Vibrational calculations for the optimized geometries were carried out to ensure that they are at energy minima. Details of calculations are included in the Appendix A. All calculations were performed with the Gaussian suit (Gaussian09, Rev. A2 and Gaussian16, Rev. C01) [23,24].

## 3. Results and Discussion

### 3.1. Assignment of Observed Vibrational Peaks

As seen in Figure 1, photoproducts varied in accordance with the UV wavelength; at λ < 290 nm, only one peak was observed at 1726 cm^−1^, and by-products such as CO_2_, CO, and OCS were found at 2343, 2138, and 2046 cm^−1^, respectively. On the other hand, at λ > 290 nm, four peaks became prominent at 947, 960, 1182, and 1195 cm^−1^. The constant intensity ratio of the three peaks at 960–1195 cm^−1^ suggested that they originate from the same species. Since a variation in the irradiation time (30 min to 1 h) did not change the spectral pattern, it is reasonable to assume that these peaks were from primary products and that the production of the secondary species was negligible. On the other hand, due to the overlap and interference with the depletion of a parental C_2_H_4_ band (ν_7_), relative intensities of the peak at 947 cm^−1^ could not be estimated quantitatively, and it was not clear whether this peak originated from the same species as those in the higher wavenumber region. Likewise, the carrier of the peak at 2920 cm^−1^ could not be identified.

At first glance, we suspected the peak at 1726 cm^−1^ to be assignable to glyoxal CHO-CHO, in analogy with the previous experiment of the C_2_H_4_-O_3_ system [21]. On the other hand, the peak at 1182 cm^−1^ seemed assignable to the ν_2_ band of dihydrogen sulfide H_2_S [25] that is produced simultaneously.
(1)$$C_{2}H_{4}+SO_{2}\overset{h\nu }{\to }CHO-CHO+H_{2}S$$

We expected that the wavelength dependence of the vibrational spectra could be accounted for by the secondary photolysis of glyoxal [26] and H_2_S. However, it turned out that the peak at 1182 cm^−1^ was too strong compared with the 1726 cm^−1^ band since the ν_2_ band of H_2_S was two orders of magnitude weaker than the C=O stretching band of glyoxal; at the B3LYP/6-31G* level infrared intensities for these bands were calculated to be 4.9 and 165 km mol^−1^, respectively, according to the CCCBDB [27]. Supplementary experiments with an ethylene isotopologue C_2_D_4_ gave us another aspect of assignments. The isotopic sample was purchased from Cambridge Isotope Laboratories (Andover, MA, USA) and used without further purification. Figure 2 shows difference spectra for the photolysis of the C_2_D_4_–SO_2_ system at 266 nm and 300 nm, with the same experimental conditions as the normal species. We found that the 1726 cm^−1^ peak was redshifted to 1688 cm^−1^ and that it was not consistent with reported infrared peak positions of fully deuterated glyoxal [28]. The spectral features in the 900–1500 cm^−1^ region showed an additional peak at 987 cm^−1^ for λ = 266 nm that could not be assigned to C_2_D_2_O_2_ either. The strong absorptions at 1182 and 1195 cm^−1^, on the other hand, remained unperturbed upon deuteration; two strong peaks were located at 1164 and 1189/1190 cm^−1^. Such behavior is not consistent with the isotopic shift of the ν_2_ band of dihydrogen sulfide. From these facts, we concluded that the photolysis of the C_2_H_4_–SO_2_ system cannot be interpreted using the mechanism (1).

Since there is no apparent mechanism that can account for the observed vibrational features, we conducted an extensive search for the photoproducts. As briefly described in the preceding chapter, 69 isomeric structures of the C_2_H_4_SO_2_ formula were constructed and subjected to DFT calculations to simulate vibrational spectra. Simulated spectra for smaller species (including sulfinic acid) did not match the observed spectral features; see Appendix A. In the first stage of this search process, we paid special attention to the photoproducts at λ > 290 nm since the vibrational spectra contain several peaks and are, thus, more informative than those in the photolysis at shorter wavelength (just one peak at 1726 cm^−1^). The candidate molecules were then limited to those without the C=O group. An initial guess for each candidate was modeled with GaussView, by considering the usual valence scheme. Figure 3 shows the top ten stable isomers obtained in the descending order of stability, and their simulated spectra are compared with the observation in Figure 4.

We found that the isomer VII (4-membered sultine, oxathietane 2-oxide) is the most plausible candidate based on a good agreement of spectral patterns with experimental observations in spite of its relatively low stability. One possible explanation would be that the other (more stable) isomers (I–V) have separated O, S, and O moieties and, thus, require large activation energy for bond breaking of SO_2_ molecule for production. As shown in Figure 5, the spectra of photoproducts in the C_2_D_4_–SO_2_ system can also be accounted for by deuterated isomer VII, and we can safely state that the photolysis of the C_2_H_4_–SO_2_ system mainly proceeds in the following scheme at λ > 290 nm.

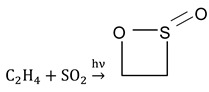
(2)


A plausible reaction scheme is proposed in Figure 6, given the production of oxathietane 2-oxide. In the electronic ground state, the C_2_H_4_–SO_2_ complex has a slipped parallel structure, as shown by FTMW studies [14,15]. Upon UV irradiation around 250–300 nm, the SO_2_ moiety is excited into the S_1_ state and promptly relaxed into the triplet state via intersystem crossings (ISC) [29,30,31,32]. The triplet SO_2_ has a biradical character and undergoes an addition reaction to the unsaturated C=C bond of ethylene. The spin density of each atom in the T_1_ SO_2_ can be estimated to be 0.85 for S and 0.57 for O, respectively, by DFT calculations at the same level of theory. The addition reaction can take place at both S and O atoms to produce two types of triplet biradical intermediates (**A** and **B**), as shown in the broken rectangle. Actually, we optimized the structure of the C_2_H_4_–SO_2_ complex at the T_1_ state and found that these two triplet biradical exist as energy minima, as shown in the inset of Figure 6. Another T-shaped isomer where SO_2_ interacts with π-electrons of ethylene directly was found rather unstable energetically. How do these intermediates relax to cyclic molecules (VI or VII)? The optimized structure of each cyclic product in Figure 6 at the T_1_ state is displayed in Figure 7. It shows that the intermediate **A** correlates to isomer VII at the T_1_ state and can relax to isomer VII via ISC and phosphorescence. Since the intermediate **A** is the most stable among the three isomeric structures in Figure 6, it seems reasonable that the photoexcitation of the C_2_H_4_–SO_2_ complex via S_1_ state preferentially leads to the isomer VII (oxathietane 2-oxide).

Another possible route for the isomer VII formation is the [2+2] cycloaddition of C_2_H_4_ and electronically excited SO_2_ (S_1_), as suggested by Salta et al. [20]. We estimated this reaction path as of minority due to the following reasons: Firstly, as we assumed in the present study, the S_1_ state is rapidly relaxed into the T_1_ state. Secondly, the [2+2] concerted reaction favors the parallel configuration of one of the S=O bonds with the C=C bond as shown below, and it is quite different from the structure of the C_2_H_4_–SO_2_ complex. It thus suggests that the cycloaddition would be slow due to the small Frank–Condon factor upon photoexcitation.

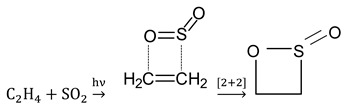
(3)


On the other hand, in the case of large excess energy (at shorter UV wavelength), it would be reasonable to assume that the triplet intermediates are subject to the S-O bond breaking followed by hydrogen migration to produce noncyclic species, as shown in the dotted rectangles of Figure 6. DFT calculations were carried out for the three potential photoproducts, and the optimized structure for each rotamer is displayed in Figure 8. Spectral matches for the photoproducts in the C_2_H_4_–SO_2_ and C_2_D_4_–SO_2_ systems in Figure 9 and Figure 10 seem favorable for the metastable isomer 1(2-hydroxysulfanylacetaldehyde) though the reproduction of the spectral features remains qualitative. The final assignments of observed peaks are listed in Table 1. Calculated frequencies for the candidate molecules obtained at the highest level of theory (B3LYP/cc-pV(T+d)Z) show a reasonable agreement with the observed peak positions after applying the appropriate scaling factor (0.965).

### 3.2. Comparison with Gas Phase Study 

From rotational analysis of REMPI action spectra, Makarov et al. concluded that molecular planes of SO_2_ and C_2_H_4_ became more parallel upon photoexcitation and that the reaction products were C_2_H_3_ + HSO_2_ [19]. Their conclusion was based on the result of an analogous system C_2_H_2_-SO_2_ [33] where an infrared emission spectrum of HSO_2_ was observed after the UV excitation of the complex and mass spectrometry. They did not mention other photoproducts, such as those in Figure 3 and Figure 8. On the other hand, we could not observe vibrational peaks of C_2_H_3_ nor HSO_2_ in the Ar matrix; spectroscopic data for these species are well documented [34,35] and can be compared with the present experimental results. These discrepancies may result from the following: Firstly, the radical pathway suggested by them is a minor reaction channel resulting from H-migration of the intermediate **A**. Secondly, these radicals originate from sequential photolysis or fragmentation during the ionization of oxathietane 2-oxide, as shown below:

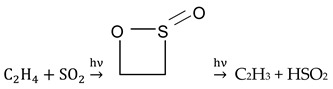
(4)


Actually, a time-dependent DFT calculation of the oxathietane 2-oxide at B3LYP-D_3_/6-31+G(d) level indicates the presence of the S_1_ state of this molecule at 280 nm above the ground state and that it is resonant with the photolysis laser. Salta et al. proposed two reaction pathways from oxathietane 2-oxide to smaller products but did not mention this radical production [20].

Another discrepancy resides in the structure of the C_2_H_4_–SO_2_ complex in excited states. The optimized structure at the T_1_ state (Figure 13 in [19]) is considerably different from those in the inset of Figure 6 though they did not give any structural parameters for that. As seen above, DFT calculations in this study are consistent with experimental results and seem more reliable.

## 4. Conclusions

As shown in the preceding sections, the photoproducts of the C_2_H_4_–SO_2_ complex in the Ar matrix upon UV excitation can be assigned to the 4-membered sultine (oxathietane 2-oxid) and a related noncyclic molecule (2-hydroxysulfanylacetaldehyde), which is consistent with the DFT calculations. More sophisticated calculations are required for the elucidation of the detailed mechanism of the photoreactions, as has been shown by Anglada et al. [36,37]. Correlated wavefunction methods, such as CASSCF, would be the way to go since the single-determinant DFT formalism cannot describe well the near-degenerate energy levels.

The influence of the matrix host on the reaction should also be addressed. One may think that the difference between the results of the present study and that of [20] originates from (at least partially) ‘cage effect’. In the present stage experimental results are consistent with DFT calculation in vacuo, and the influence of cage effect seems negligible. Similar experiments in other matrices (Ne and parahydrogen) would shed light on the influence of the cage effect on the photoreaction of the C_2_H_4_–SO_2_ system.

Finally, we would like to point out different behaviors upon photoexcitation between the C_2_H_4_–SO_2_ system and the C_2_H_4_–O_3_ system [21] in spite of the isoelectronicity of the two oxidant molecules in the valence shell. The branching ratio of the two reaction paths leading to the 4-membered sultine and the noncyclic aldehyde in the former system shows remarkable and seamless UV wavelength dependence. On the other hand, photolysis in the C_2_H_4_-O_3_ system does not show such dependence: two reaction products (ethylene oxide and glyoxal) coexist throughout the 300–670 nm region [21]. These differences may be (at least in part) correlated to the difference of the nonadiabatic couplings in these molecules; spin-orbit coupling plays an important role in SO_2_ [29], whereas the vibronic couplings are prominent in O_3_. Such subtle differences in the primary photoreactions may lead to a variety of SOAs with different compositions/properties originating from these two oxidant molecules in the atmosphere.

## Figures and Tables

**Figure 1 molecules-29-05362-f001:**
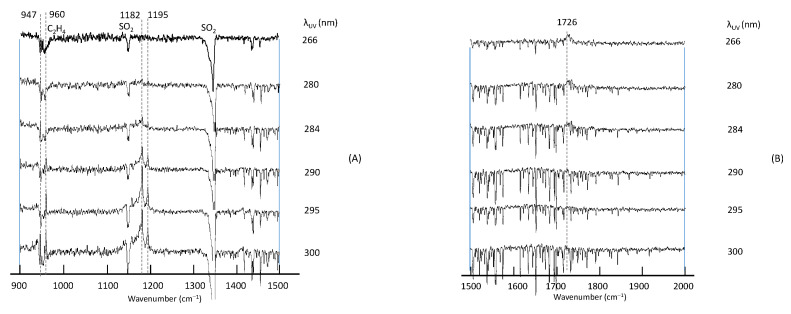
Difference spectra of the C_2_H_4_–SO_2_ system in Ar matrix after UV excitation in (**A**) 900–1500 cm^−1^, (**B**) 1500–2000 cm^−1^, (**C**) 2000–2500 cm^−1^ and (**D**) 2500–3000 cm^−1^ regions.

**Figure 2 molecules-29-05362-f002:**
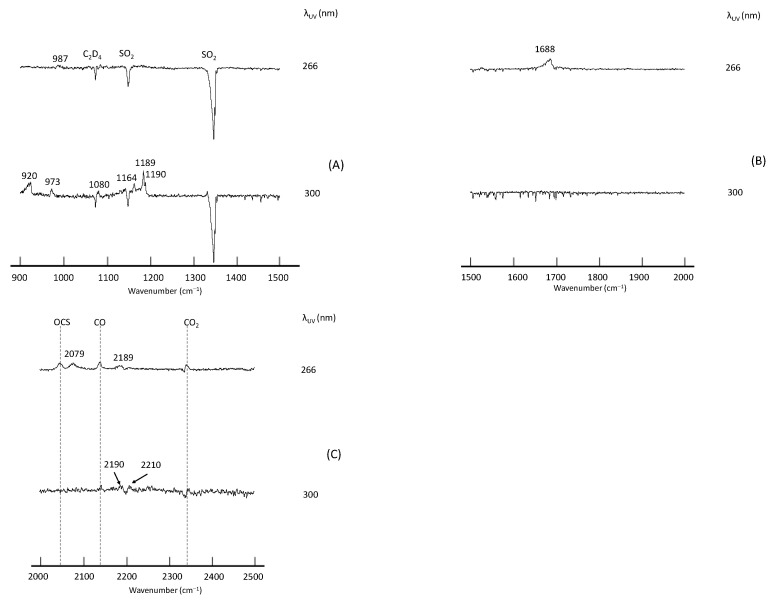
Difference spectra of the C_2_D_4_–SO_2_ system in Ar matrix after UV excitation in (**A**) 900–1500 cm^−1^, (**B**) 1500–2000 cm^−1^ and (**C**) 2000–2500 cm^−1^ regions.

**Figure 3 molecules-29-05362-f003:**
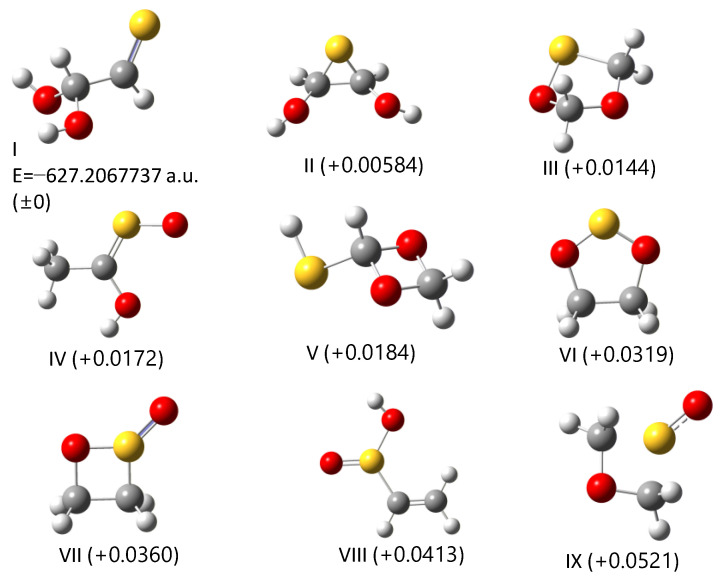
Stable isomers with the composition formula C_2_H_4_SO_2_ obtained from DFT calculations at the B3LYP/6-31+G(d) level. Numbers in parentheses are relative energy to the most stable isomer I in a.u.

**Figure 4 molecules-29-05362-f004:**
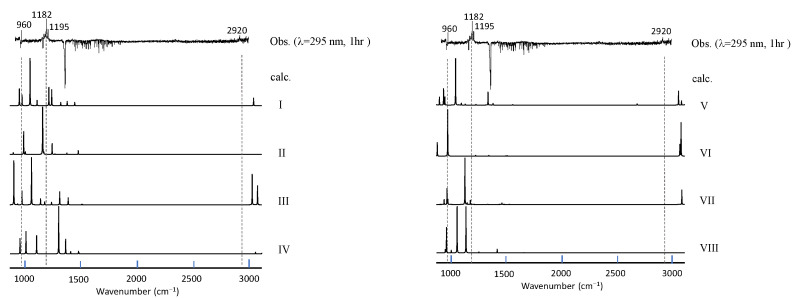
A comparison of observed vibrational spectra of photoproducts in the C_2_H_4_–SO_2_ system at longer UV wavelength with a simulation for each isomer (I to VIII in Figure 3). No scaling factor is applied.

**Figure 5 molecules-29-05362-f005:**
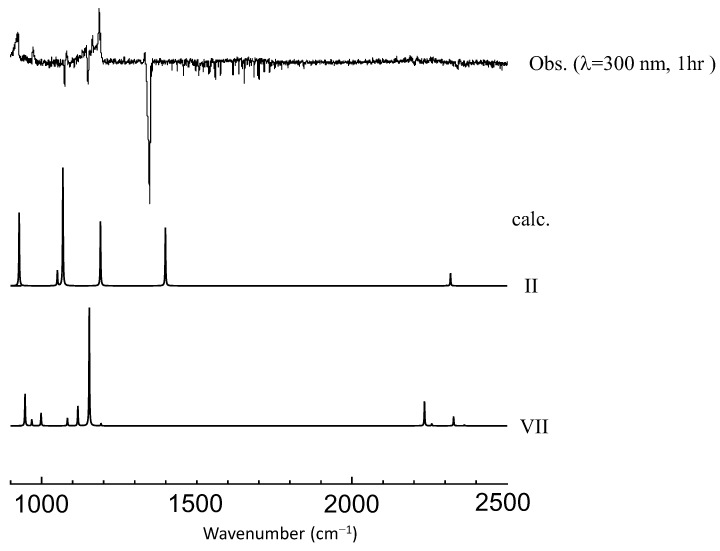
A comparison of observed vibrational spectra of photoproducts in the C_2_D_4_–SO_2_ system at longer UV wavelength with a simulation for each isomer (II and VII in Figure 3). No scaling factor is applied.

**Figure 6 molecules-29-05362-f006:**
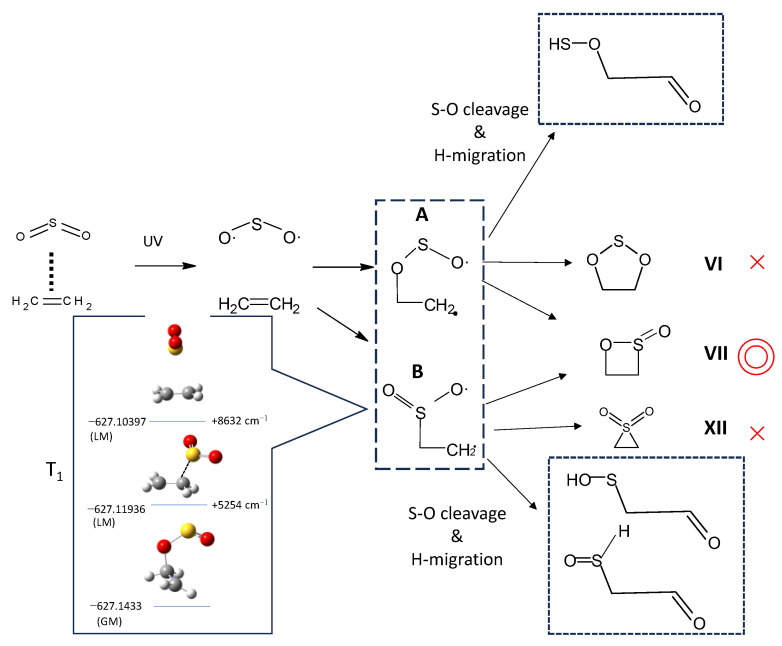
A plausible reaction mechanism for the C_2_H_4_–SO_2_ system after UV excitation. An inset shows the stable isomers of C_2_H_4_SO_2_ in the lowest triplet states (T_1_) obtained from DFT calculations at the B3LYP-D_3_/6-31+G(d) level.

**Figure 7 molecules-29-05362-f007:**
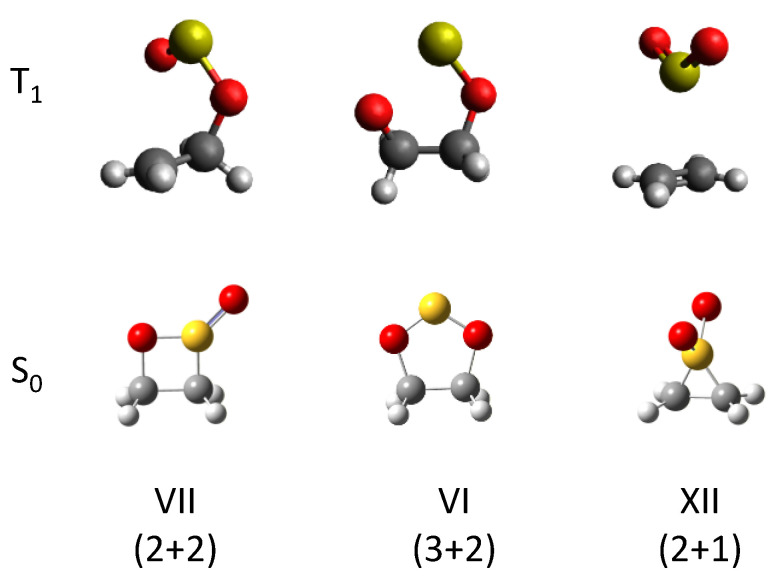
Optimized structures for the ring-type stable isomers of C_2_H_4_SO_2_ in S_0_ and T_1_ states at the B3LYP-D_3_/6-31+G(d) level.

**Figure 8 molecules-29-05362-f008:**
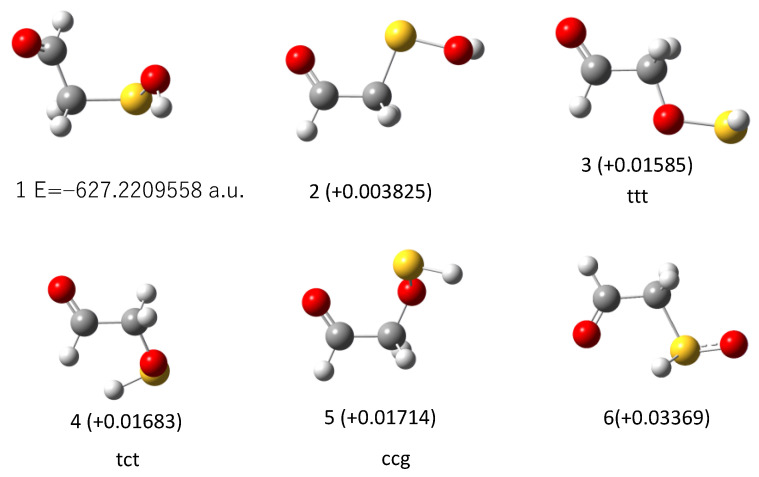
Candidate molecules for the photoproducts at shorter UV wavelength according to the reaction scheme in Figure 6. Structural optimization for each isomer was performed at the B3LYP/6-31+G(d) level.

**Figure 9 molecules-29-05362-f009:**
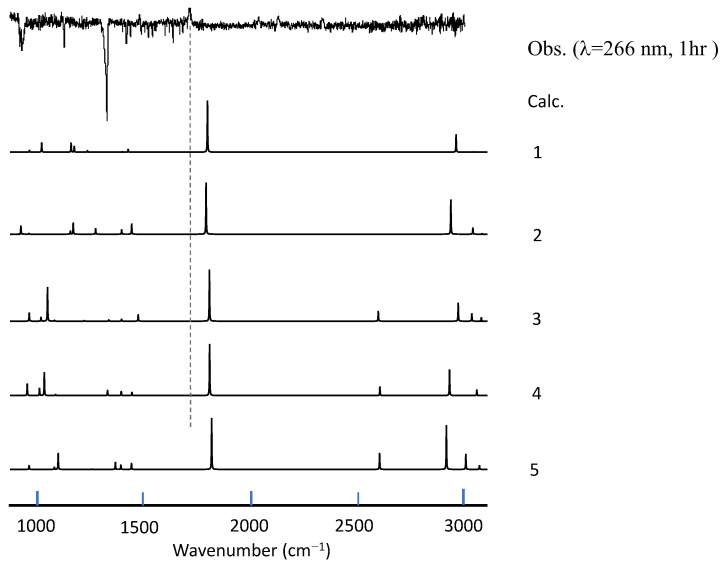
A comparison of observed vibrational spectra of photoproducts in the C_2_H_4_–SO_2_ system at shorter UV wavelength with a simulation for each isomer in Figure 8. No scaling factor is applied.

**Figure 10 molecules-29-05362-f010:**
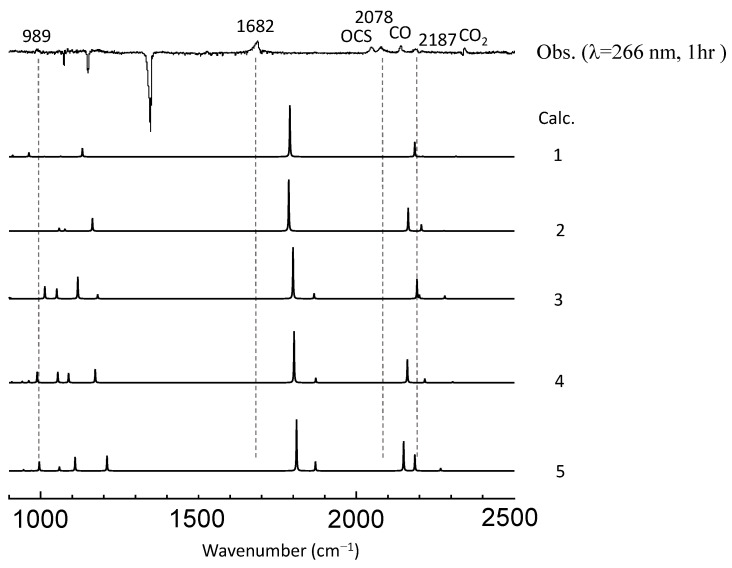
A comparison of observed vibrational spectra of photoproducts in the C_2_D_4_–SO_2_ system at shorter UV wavelength with a simulation for each isomer in Figure 8. No scaling factor is applied.

**Table 1 molecules-29-05362-t001:** Observed vibrational peak positions and assignments.

Observed Peak Position (cm^−1^)	Calculated Peak Position (cm^−1^) ^a^	Assignment	Vibrational Mode
C_2_H_4_–SO_2_			
947	919	oxathietane 2-oxide	CH_2_ twist
960	950	oxathietane 2-oxide	C-O str
1182	1170	oxathietane 2-oxide	S=O str + CH_2_ twist
1195	1177	oxathietane 2-oxide	S=O str + CH_2_ twist
1726	1745	2-hydroxysulfanylacetaldehyde	C=O str
2920	2945	oxathietane 2-oxide	CH_2_ str
C_2_D_4_–SO_2_			
920	909	oxathietane 2-oxide-d_4_	C-O str
973	949	oxathietane 2-oxide-d_4_	CD_2_ wag
987	924	2-hydroxysulfanylacetaldehyde-d4	CD_2_ wag
1080	1068	oxathietane 2-oxide-d_4_	CD_2_ wag
1164	1132	oxathietane 2-oxide-d_4_	C-C str
1189	1173	oxathietane 2-oxide-d_4_	S=O str
1190		oxathietane 2-oxide-d_4_	Fermi? ^b^
1688	1724	2-hydroxysulfanylacetaldehyde-d_4_	C=O str
2079	2070	2-hydroxysulfanylacetaldehyde-4	C-D str
2189		2-hydroxysulfanylacetaldehyde-d_4_	Fermi? ^b^
2190	2139	oxathietane 2-oxide-d_4_	CD_2_ sym str
2210	2226	oxathietane 2-oxide-d_4_	CD_2_ asym str

^a.^ scaled wavenumber (factor 0.965) from the results at B3LYP/cc-pV(T+d)Z level of theory. ^b.^ These transitions cannot be assigned to fundamental bands and are tentatively assigned to overtone/combination band borrowing intensity via Fermi resonance.

## Data Availability

Data are contained within the article and Appendix A.

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
