# Peer review of "Photoreactions of the C2H4–SO2 Complex in a Low-Temperature Matrix Investigated by Infrared Spectroscopy and Density Functional Theory Calculations"

_molecules, 2024, doi:10.3390/molecules29225362_

Round 1
Reviewer 1 Report
Comments and Suggestions for Authors
This paper is well presented and a useful addition to the the literature of the reaction of small molecules found in combustion situations. The work is well summarized and expanded by a Conclusion section. The English is quite good and does not present a problem.
One section needs attention is the Abstract which I found a little confusing.
Rather than going through all the issues point by point, I have given my rewrite of the Abstract as:
"Abstract: Ethylene and sulfur dioxide molecules were co-deposited on a CsI window at cryogenic temperature, and the photoproducts upon UV irradiation were observed using Fourier-transformed infrared (FTIR) spectroscopy. The products were found to be UV wavelength-dependent; at shorter wavelengths (λ=266 nm) one strong peak was observed while more than three peaks were identified at longer UV wavelengths (λ=300 nm). Spectral features changed seamlessly along with UV wavelength. Density functional theory (DFT) calculations were carried out for potential products, and spectral matches between observations and calculations seemed satisfactory, assuming a cyclic molecule (oxathietane 2-oxide) as the main photoproduct at longer UV wavelengths. On the other hand, the spectra of photoproducts at shorter UV wavelengths were reproduced by assuming the decomposition products of an intermediate, from the supplementary experiments using deuterated samples. Plausible photoreaction schemes were presented to account for the observed photoproducts."
With these small changes, the paper is acceptable for publication.
Author Response
The authors would like to show sincere gratitude to the positive comments from the reviewer. The abstract is rewritten accordingly.
Reviewer 2 Report
Comments and Suggestions for Authors
The manuscript describes a combined experimental-theoretical study of the photoreaction of SOâ‚‚ with ethylene. The experimental and theoretical methods are appropriate for this investigation, and both the methodology and conclusions appear sound. My only observation concerns the proposed mechanism of intermediate VII formation. Why did the authors consider only a non-concerted mechanism? It is well known that the 2+2 photoaddition of ethylene proceeds through a concerted process. It would be valuable to calculate the transition states for both possible mechanisms to determine which is more favorable.
Author Response
The authors really appreciate valuable comments from the reviewer. The [2+2] cycloaddition was not taken into account in this study, since this reaction takes place in the S1 state, whereas in this study we assumed that SO2 in the T1 state undergoes reaction with C2H4, due to fast intersystem crossing from S1 to T1. Another reason was that the [2+2] concerted reaction favors parallel configuration of one of the S=O bonds and the C=C bond and it is quite different from the structure of the C2H4--SO2 complex. Therefore, it seems that the cycloaddition reaction would be very slow due to the small Frank-Condon factor upon photoexcitation.
Reviewer 3 Report
Comments and Suggestions for Authors
The paper presents a study of the photoreaction of C$_2$H$_4$–SO$_2$ complex. Sulfur chemistry is a critical topic in theoretical, physical and environmental chemistry and thus will be of interest to many readers. The paper is well-written and I very much enjoy reading the paper. I suggest the authors to make some revision to address my questions.
1. Can the author comment on the usage of different electronic structure program for this system? The excited states (both with singlets and triplets) must be calculated with some dft variants such as tddft. It would be nice to know.
The system size probably sits at the intersection of dft and wavefunction methods. Would the use of wavefunction based methods, such as CAS methods improve the results? Photoreactions can depend on the description of near-degenerate states, i.e. conical intersections, would dft describe that well?
2. This is related to the previous questions. In line 173, the author says SO$_2$ is promptly relaxed back to T$_1$ state, which is a statement that could be right depend on the context. Do the authors have a estimation on the spin orbit coupling values? The reaction activity difference between SO$_2$ and O$_3$ is largely dependent on the spin-orbit effect. Is that taken into consideration in the comparison?
3. Line 34, olefines to olefins.
Line 182, is the sentence with question mark a undelete comment?
Author Response
The authors really appreciate valuable comments from the reviewer. Responses to the comments are listed below.
(1) As experimentalists, we are concentrated in the assignments of observed vibrational spectra and a reasonable interpretation of the photoproducts. For this purpose, DFT calculations with the Gaussian suites are the first choice. We have used CP2k and NWChem for supplementary calculations. During the course of this study, we made some calculations on the C2H4--SO2 complex in the S1 state with TDDFT formalism, but we could not get positive results due to the presence of ghost states. The employment of another functional wB98xD did not improve the situation. So we did not include the results of these calculations in the manuscript. As pointed by the reviewer, it is known that (single-determinant) DFT cannot describe the near degenerate states in the conical intersections. In that sense, in order to describe the detailed mechanism of the photoreaction of the present system, correlated wavefunction methods, such as CASSCF, would be the way to go, although it is not an easy way for the experimentalists like us. Another pathway would be to use spin-flip DFT to get better description of the degenerate systems. Since this functionality is not implemented in the Gaussian and other software that I used to, I will learn how to use ORCA or GAMESS. These points are included in the Conclusions.
(2) We must confess that we did not estimate the spin-orbit coupling of SO2, simply because the Gaussian does not offer such functionality. We tried to calculation the SOC by using NWChem, but it was not successful; available information is quite limited for this software. As pointed out by the reviewer, the SOC plays an important role in the non-adiabatic couplings of SO2; in case O3 SOC is much smaller and vibronic couplings would be prominent. Such comparison is included in the Conclusions.
(3) Our apology to the typographical error and a bad hyphenation. They are corrected in the revised manuscript.